# Comparison of Study Quality as Determined by Standard Research and Community Engagement Metrics: A Pilot Study on Breast Cancer Research in Urban, Rural, and Remote Indigenous Communities

**DOI:** 10.3390/ijerph19095008

**Published:** 2022-04-20

**Authors:** Vita Christie, Janaki Amin, John Skinner, Debbie Green, Karen Littlejohn, Kylie Gwynne

**Affiliations:** 1Faculty of Medicine Health and Human Sciences, Macquarie University, Sydney, NSW 2109, Australia; janaki.amin@mq.edu.au (J.A.); kylie.gwynne@mq.edu.au (K.G.); 2Poche Centre for Indigenous Health, The University of Sydney, Sydney, NSW 2006, Australia; john.skinner@sydney.edu.au; 3Armajun Aboriginal Health Service, Armidale, NSW 2350, Australia; dgreen@armajun.org.au; 4Foundation for Breast Cancer Care, South Brisbane, QLD 4101, Australia; littlejohnkaren67@gmail.com

**Keywords:** research methodologies, indigenous health, health knowledge, attitudes, practice, community participation

## Abstract

The purpose of this review is to compare research evaluation tools to determine whether the tools typically used for assessing the quality of research adequately address issues of Indigenous health and culture, particularly when the studies are intended to benefit Indigenous peoples in urban, regional, rural, and remote settings. Our previously published systematic review evaluated studies about breast cancer using a modified Indigenous community engagement tool (CET). In this study, we evaluated the same studies using two commonly used tools: the Critical Appraisal Skills Programme (CASP) for qualitative research; and the Effective Public Health Practice Project (EPHPP) for quantitative research. The results were then compared to ascertain whether there was alignment between performances in terms of engagement and the CASP/EPHPP metrics. Of the 15 papers, 3 papers scored weakly on both metrics, and are therefore the least likely to offer reliable findings, while 2 papers scored strongly on both metrics, and are therefore the most likely to offer reliable findings. Beyond this summation, it was clear that the results did not align and, therefore, could not be used interchangeably when applied to research findings intended to benefit Indigenous peoples. There does not appear to be a pattern in the relationship between the reliability of the studies and the study settings. In order to address disparities in health outcomes, we must assess research through a typical research quality and cultural engagement and settings lens, ensuring that there is rigour in all aspects of the studies.

## 1. Introduction

A systematic review is a review of a clearly formulated question that uses systematic and reproducible methods to identify, select, and critically appraise all relevant research, and to collect and analyse data from the studies that are included in the review [1]. By looking at and evaluating a large number of comparable studies, a systematic review can provide answers that have a much stronger level of evidence than any individual study. Evidence from systematic reviews can be used to inform healthcare program design and evaluation, and it is often assumed that evidence found to be strong through a systematic review will play out in the same way for Indigenous peoples (across various settings, e.g., urban, regional, rural, and remote) as it does for the wider population. This is not necessarily the case, and tools used to assess the quality of evidence do not typically assess cultural or contextual factors related to that research.

We know that research and services for Indigenous peoples must be conducted using culturally safe and contextualised methods that are also scientifically rigorous if their findings are to be appropriate, reliable, and replicable. Cultural safety recognises power differentials in healthcare settings, and how the transfer of power from health professionals to those being cared for can facilitate appropriate care for Indigenous people [2] Many Indigenous studies are small-scale and/or qualitative [3]. When conducting a systematic review that looked at what the global evidence was for culturally safe strategies to improve breast cancer outcomes for Indigenous women in high-income countries, the authors [4] utilised an evaluation tool that focused specifically on ethical research with Indigenous communities, and was based on the guidelines set forth by the Australian National Health and Medical Research Council [5]—a modified community engagement tool (CET). The CET was developed by the authors to provide an overall score of community engagement in the research. The tool was then implemented, privileging the perspectives of Indigenous investigators.

Typical research evaluation tools—which assess study quality—focus on the reliability and replicability of the research method, and do not assess metrics related to cultural safety. Typical research evaluation tools used in systematic reviews to evaluate the quality and reliability of evidence [6] include the Critical Appraisal Skills Program (CASP; Oxford, UK [7]) for qualitative research, and the Effective Public Health Practice Project (EPHPP; Toronto, ON, Canada [8]) for quantitative research. They do not include any measures of community engagement, cultural safety, or setting. A different approach that decolonises the research process and interprets its findings through a cultural lens is required to identify evidence that is genuinely appropriate, reliable, and replicable for Indigenous peoples.

This review examines two types of research evaluation tools and the ways in which they rate the quality of studies and concordance with the assessment of culturally reliable evidence in healthcare research. The study provides an example of a way to identify the best possible evidence—in this case, for improving breast cancer treatment and outcomes for Indigenous women in various geographical settings in high-income countries.

## 2. Method

We reassessed the studies selected from our previously published systematic review [4], using the PRISMA guidelines for peer-reviewed articles in the Medline, EMBASE, CINAHL, Scopus, Web of Science, ProQuest Sociology, and Informit Rural health databases and Indigenous collection databases. Key inclusion criteria were as follows: adult female patients with breast cancer; high-income country setting; outcome measures were uptake and level of satisfaction for women, including screening, diagnosis, treatment, and follow-up care. The tool used for the systematic review was an adaptation of the Australian National Health and Medical Research Council’s Guidelines for Ethical Conduct in Aboriginal and Torres Strait Islander Health Research [5]—a community engagement tool (CET).

The papers were rated as 1 for yes and 0 for no on each of five criteria: (1) issue identified by community; (2) Indigenous governance; (3) capacity building; (4) cultural consideration in design; and (5) respecting community experience. All papers were scored by V.C., K.G., and D.G. Each paper was then categorised as strong (score of 4 or 5), moderate (2 or 3), or Weak (0 or 1). This method was reported previously [4].

Of the 15 papers included in the systematic review, there were 8 qualitative and 7 quantitative papers. The qualitative papers were assessed independently using the CASP by two authors (V.C. and K.G.). The quantitative papers were assessed using the EPHPP by three authors (J.A., V.C., and K.G.).

With all three tools (the CET, CASP, and EPHPP), the authors rated the papers independently, and where there were differences, they were resolved by discussion and consensus, privileging the opinion of the Aboriginal author (D.G.) when applying the CET. The privileging of Indigenous voices is an accepted research methodology [6]. The scores were then collated and compared to capture the frequency of different combinations of scores. The similarities and differences between the tools were then analysed.

Full analysis of the text was undertaken by two authors (V.C. and K.G.), and all data—including geographical settings of urban, regional, rural, and remote—were recorded in an Excel spreadsheet (Table 1).

## 3. Results

A record of the scores and settings for each paper is shown in Table 1. The frequency of scores is shown in Table 2. Of the nine papers that scored as strong on the CET, two scored as strong on the CASP/EPHPP; of the three that scored as strong on the CASP/EPHPP, two scored high on the CET. Three papers scored as weak on both. Two papers scored as strong on both metrics. None of the papers reported their results by geographical setting, even when most (*n* = 13) specifically recruited participants and/or collected that demographic information.

Table 3 describes the characteristics of papers in each of the categories and identifies why papers scored as strong or weak in each evaluation.

## 4. Discussion

Our study found that in the context of studies examining strategies to improve breast cancer outcomes for Indigenous women in high-income countries it is important to apply both the CET and the CASP/EPHPP. In evaluating the value of research that is intended to benefit Indigenous peoples, a cultural evaluation is necessary to determine the reliability and cultural safety of that research. However, cultural tools alone cannot determine the replicability and validity of interventions and should be combined with tools that evaluate research methods and the validity of study design. An example of this is shown in the EPHPP automatically considering randomised control trials and controlled clinical trials to be strong in terms of study design, whereas these types of studies often do not translate effectively to Indigenous research. Community engagement is about ethical research from a cultural perspective, and while ethics is included as a measure within the qualitative and quantitative evaluation tools, culture is not explicitly addressed.

The authors closely examined the studies that scored “strong” across both the community engagement and the typical tools and were able to identify elements that they had in common, indicating the elements that are workable in both contexts and that strengthen a study. In such studies, the team found some generalisable findings that are reliable through the cultural and community lens, along with methodologies that were repeatable. It was also noted that the studies that scored highly in both categories had reported high levels of detail for them to be well planned and organised studies with considered methodologies, whilst ethical and cultural focus was maintained.

Upon analysis of the texts, while most (*n* = 13) collected data on geographical setting, none of the papers reported the setting in their results. This is despite some studies (*n* = 4) specifically recruiting from urban and rural/remote settings. The compounding impact of distance, local access to services and skilled workforce, and lack of reliable transportation on health outcomes for people living outside of urban areas has been widely reported in the health services literature. It is possible that Indigenous women are so disadvantaged in access to and participation in breast cancer prevention and treatment services that significant differences cannot be seen in this population based on setting. Further research is warranted to explore this, as the authors contend that rurality and indigeneity are compounded in outcomes data related to breast cancer.

## 5. Conclusions

Closing the gap in health outcomes between Indigenous and non-Indigenous peoples will require Indigenous governance and leadership, as well as rigorous study design methodologies and sufficient sample sizes. It is not appropriate to rely on evidence intended for Indigenous peoples that has been developed without Indigenous engagement, governance, and control. This study provides a way to bring together two types of research analysis that, when used together, make it more likely that research is rigorous, reliable, and culturally safe. The two examples in this paper of studies with “strong” scores on both tools provide a sound basis for culturally safe approaches to improving breast cancer prevention and treatment services.

### Limitations

This pilot study utilises only two typical research analysis tools. It is unclear whether the findings of this study apply to all tools used for research analysis, and whether these findings are generalisable beyond breast cancer, indigenous health, and high-income settings. Further research is warranted to explore the benefits of combining tools that assess the cultural value of the research as well as the sample size, replicability, and generalisability of the studies. Due to its comparison with a previously published systematic review, there was only a small sample of studies available for this review.

## Figures and Tables

**Table 1 ijerph-19-05008-t001:** Results of the community engagement tool (CET), typical tool scores, and setting by paper.

Papers	CET	EPHPP/CASP	Qualitative or Quantitative	Setting
Becker et al. [9]	Strong	Strong	Qualitative	Not reported
Strickland et al. [10]	Strong	Strong	Qualitative	Not reported
Daley et al. [11]	Strong	Moderate	Qualitative	Rural/urban
Sinicrope et al. [12]	Strong	Moderate	Quantitative	Urban/rural
Banner et al. [13]	Strong	Weak	Quantitative	Rural
Brown et al. [14]	Strong	Weak	Quantitative	Rural
Daley et al. [15]	Strong	Weak	Qualitative	Rural/urban
English et al. [16]	Strong	Weak	Quantitative	Rural/remote
Ka’opua et al. [17]	Strong	Weak	Quantitative	Rural
Pilkington et al. [18]	Moderate	Strong	Qualitative	Urban/regional/rural/remote
Haozous et al. [19]	Moderate	Moderate	Qualitative	Regional
Sanderson et al. [20]	Weak	Moderate	Qualitative	Rural/remote
Friedman et al. [21]	Weak	Weak	Qualitative	Urban
Roh et al. [22]	Weak	Weak	Quantitative	Urban
Tolma et al. [23]	Weak	Weak	Quantitative	Rural

**Table 2 ijerph-19-05008-t002:** Frequency of scores.

CASP or EPHPP	CET Scoring		
	Strong	Moderate	Weak
Strong	2	1	0
Moderate	2	1	1
Weak	5	0	4

**Table 3 ijerph-19-05008-t003:** Frequency and summary of analysis.

Community Engagement Tool/CASP or EPHPP	Frequency	Analysis
Strong/strong	2	Both studies are qualitative; adequate attention to detail around ethical considerations; both the CASP and CET focus on research according to the needs of the group being researched, with the CET looking specifically at the needs of Indigenous peoples
Strong/moderate	2	Strong Indigenous governance in research design and reporting; very clear description of eligibility criteria; small sample size
Strong/weak	5	Strong Indigenous engagement in design and reporting; focused on working in appropriate contexts with appropriate planning and consultation prior to study; both cohort studies, and both lacking detail regarding one group; majority in this category are quantitative
Moderate/strong	1	Research not led by the Indigenous community, but otherwise the needs of the community are adequately covered; clear and appropriate recruitment strategy and rigorous data collection and analysis
Moderate/moderate	1	Small sample size; methodology was not clear regarding design, recruitment strategy, data collection, or relationship of researcher to participants.
Moderate/weak	0	
Weak/strong	0	
Weak/moderate	1	Unsuitable methodology (some interviews self-administered); small sample size; limitations outweigh benefits
Weak/weak	3	No Indigenous governance; small sample size; research design and data collection did not suit research questions; data analysis not rigorous

## Data Availability

No datasets were generated or analysed during this study.

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
