# Peer review of "Comparison of Study Quality as Determined by Standard Research and Community Engagement Metrics: A Pilot Study on Breast Cancer Research in Urban, Rural, and Remote Indigenous Communities"

_ijerph, 2022, doi:10.3390/ijerph19095008_

Round 1

Reviewer 1 Report

This sysematic review is a small compact review paper, which is assessed using proper methods breast cancer research in urban, rural and remote indigenous communities. In publications, which are of great importance for setting and prevention, quality, reproducibility and comparable data are important.

In the metods, I would also like to read how the publications were selected, the time period, not only how they were evaluated.

In table no. 1, I would like to see the year of publication next to the authors and title.

Author Response

Section

Reviewer comment

Authors’ response

General

This systematic review is a small compact review paper, which is assessed using proper methods breast cancer research in urban, rural and remote indigenous communities. In publications, which are of great importance for setting and prevention, quality, reproducibility and comparable data are important.

Thank you for this observation, we agree.

Methods

In the methods, I would also like to read how the publications were selected, the time period, not only how they were evaluated.

Thank you for this suggestion, we have added that detail to the text.

Lines: 90-96

Tables

In table no. 1, I would like to see the year of publication next to the authors and title.

Thank you for this suggestion, these have been added to Table 1.

Reviewer 2 Report

To summarize I think that the topic is important and of interest to readers who do Indigenous health research. I think that the manuscript could be improved by a bit more elaboration regarding the difference between various systematic reviews and metasynthesis as well as a line or two about realist synthesis. It was unclear to me if the authors were attempting to fill a gap or highlight that there is a gap in tools available to assess Indigenous Health research. It would be ideal if that was clear.

  1. I’m not sure what is meant by “we assume that evidence found to be strong through a systematic review will play out in the same way for Indigenous peoples as it does for the wider population”. I’m sure this is true for some research questions (Do triple negative patients benefit from neoadjuvant chemotherapy vs post-operative adjuvant chemotherapy) whereas not for other research questions (is neoadjuvant chemotherapy the best treatment option for Indigenous patients with triple negative breast cancer).
  2. The systematic review they conducted (what is the global evidence for culturally safe strategies to improve breast cancer outcomes for Indigenous women in high income countries) asks a complex question and does not meet the definition of a review of a clearly formulated question. The type of systematic review given the available literature and the question posed would be important and one might consider a realist synthesis in this situation to get at the understanding of what worked in what context and why rather than if a particular intervention works or not. This would allow for culture and community engagement to be considered as a context.
  3. Agree that typical tools used to evaluate quality of research articles do not include any measure of community engagement, cultural safety or setting and that this represents a gap that needs to be filled. It is not clear to me if this paper is intended to close this gap or further document the need.

Author Response

Section

Reviewer comment

Author response

To summarize I think that the topic is important and of interest to readers who do Indigenous health research. I think that the manuscript could be improved by a bit more elaboration regarding the difference between various systematic reviews and metasynthesis as well as a line or two about realist synthesis.

 It was unclear to me if the authors were attempting to fill a gap or highlight that there is a gap in tools available to assess Indigenous Health research. It would be ideal if that was clear.

Thank you for your comments. The paper is about highlighting a gap and offering one way of addressing it. We have added this to the introduction.

Lines: 50-52

The paper is not a realist review nor does it include metasynthesis.

I’m not sure what is meant by “we assume that evidence found to be strong through a systematic review will play out in the same way for Indigenous peoples as it does for the wider population”. I’m sure this is true for some research questions (Do triple negative patients benefit from neoadjuvant chemotherapy vs post-operative adjuvant chemotherapy) whereas not for other research questions (is neoadjuvant chemotherapy the best treatment option for Indigenous patients with triple negative breast cancer).

Thank you for the comment, we have reworded this to be clear about the intended purpose of the systematic review in relation to this research.

Line: 46

We did not intend to generalize to all systematic reviews.

The systematic review they conducted (what is the global evidence for culturally safe strategies to improve breast cancer outcomes for Indigenous women in high income countries) asks a complex question and does not meet the definition of a review of a clearly formulated question. The type of systematic review given the available literature and the question posed would be important and one might consider a realist synthesis in this situation to get at the understanding of what worked in what context and why rather than if a particular intervention works or not. This would allow for culture and community engagement to be considered as a context.

Thank you for this observation. We agree a realist review of the limited evidence may be of value, however, for our systematic review (now published) we adopted thematic analysis using inductive coding. Importantly, our methods privileged the perspectives of Indigenous investigators.

Agree that typical tools used to evaluate quality of research articles do not include any measure of community engagement, cultural safety or setting and that this represents a gap that needs to be filled. It is not clear to me if this paper is intended to close this gap or further document the need.

Thank you for this comment.

The paper is about highlighting a gap and offering one way of addressing it. We have added this to the introduction.

Lines: 77-79

Reviewer 3 Report

Thank you for the opportunity to review this paper.

There is great need to critically review the tools being used to assess quality of Indigenous studies and papers.

I have made a number of suggestions to help improve the reader’s understanding of a number of aspects in this paper including:

  • The CET tool – exactly what is it? What does it contain? Did this team modify it or was it already modified?
  • Suggested changes, addition of references and deeper discussion about terminology used – for example how can concepts of cultural safety, community settings, and reliability best be used together?

Underlying all of this research, the tools and appraisal, is a background of colonisation and decolonisation of Western health systems, health care and the way research is conducted and reported. Each country is in a different phase of addressing these aspects, and so identifying the country of origin of the papers and the tools is also important, particularly in an international context.

Title

This paper focuses more on the comparison of study quality, than the topic of Breast cancer – I suggest changing the order of the two parts of the title to reflect this

 Comparison of study quality as determined by 3 standard research and community engagement metrics in breast cancer research in urban, rural and remote indigenous 2 communities.

Terminology in abstract and main body

Use of word ‘typical’ to describe the tools. I suggest using another word such as ‘often’.  This enables a deeper consideration of whether the ‘typical’ tools used by Western research are appropriate and inclusive of Indigenous preferred approaches and priorities. ‘Typical’ seems exclusive of these considerations.

The modified Indigenous community engagement tool – suggest put (CET) following in the abstract as this is used throughout the article. Also did you modify this tool, or did you use an already modified version? This is unclear.

Reliability – this term is used throughout the paper. Can the authors explore/discuss more deeply the link and considerations between reliability as a quality measure, and recognising and responding to individual community needs and context as described in cultural safety.  Also, are Western quant methods ‘reliable’ from an Indigenous community perspective? To what extent have Aboriginal/Indigenous peoples been involved in critically reviewing the underlying assumptions and therefore the ‘reliability’ of these tools, from their perspective? This is an emerging discussion amongst many Indigenous and non-Indigenous peoples internationally.

The authors use cultural safety as a concept throughout, but have not offered a reference to identify which definition of cultural safety they are referring to. Please include.

There also seems to be a link being made between cultural safety and community settings and at times they are almost used interchangeably – Table 1 identifies setting, but nothing else about cultural safety - can this be explained more clearly.

The tools

The CET tool – I am unclear exactly what the CET is. Having looked at reference 4 – NHMRC I am still confused as to exactly what the CET tool is, the best reference to find it, and whether this authorship team have modified it, or whether it has already been modified. This makes it difficult to assess and understand more deeply the connections being made between cultural safety, setting, and cultural engagement. Also whether Indigenous Governance is part of this assessment. The importance of Indigenous Governance is introduced in the discussion section. I suggest also include in the Introduction (briefly).

Suggest authors use a reference for each tool that enables the reader to go straight to the tool in question

Also include the CET tool as a figure or table. Without this, it is hard to really review this paper and the discussion and conclusions effectively.

These tools are also from different countries, please identify country of origin.  There may be contextual history and colonisation/decolonisation context behind each tool.

The authors may also be interested in looking at the CREATE tool which is highly relevant. https://bmcmedresmethodol.biomedcentral.com/articles/10.1186/s12874-020-00959-3

I am not suggesting that you need to use this tool for this review and paper, (being that you have already completed the review) but you may find it useful in your next work. We are finding it increasingly useful, as it brings together many of the concepts you are discussing in this paper.

Objective: To develop an instrument that appraises the ethical and methodological quality of research conducted with Aboriginal and Torres Strait Islander peoples from an Aboriginal and Torres Strait Islander perspective.

The instrument is designed to be use used in conjunction with existing critical appraisal tools.

Introduction

P1 line 36 –38 “We assume that”…suggest rephrasing, it is unclear whether this is your assumption, or whether you are questioning this assumption (hopefully the latter). This assumption is pretty colonial if using Western tools. 

P2 line 46 – this is where the reference clearly identifying the CET tool origin is needed.

The importance of Indigenous Governance is discussed for the first time in the discussion. It should be introduced here also.

P2 line 48 – ‘Typical’ suggest Western or similar term- as discussed in my terminology comments.

P2 line 49- cultural safety – please provided some examples of what you mean by cultural safety and/or a reference

P2 –line 54 – a different approach is needed, rather than may be. Suggest you could use stronger language.

Method

Suggest rewording first sentence. We reassessed the studies selected…[or similar wording] 

It’s good that you have included that the authorship includes one Aboriginal author. This information is important.

Results

Tables

The tables could benefit by some formatting/structural changes. It may also be that they were harder to read spanning across pages.

Headings- use same terminology for tools in all 3 tables to reduce confusion to readers new to the terms

Table 1 – ‘Typical’ tools – do you mean CASP/EPHPP – if so, use those terms

Table 3- suggest used CET/ CADSP or EPIHPP

Table 1 – suggest format each reference detail to left to make it easier to read. I note that these references are not included in the reference list, nor is the date of publication of these papers.   Perhaps include dates or include in reference lists, and/or refer to your original paper.  The dates are relevant as papers tend to improve quality over time. It may also be helpful to indicate which country each paper is from, as there may be trends as different countries respond to quality in writing Indigenous papers. 

Discussion

Line 99 – the value of research – to whom?

What does a cultural evaluation of reliability mean or look like? Is it that the study reliably works with community members? Or is reliability referring to quantitative studies?

Line 103 Research method precision – what does this mean for the qualitative studies? Or is that sentence referring to quant and then qual studies?

Second paragraph – can the authors unpack a little more the generalizable findings, that are reliable through a cultural and community lens, and methodologies that are repeatable, when cultural safety is also about responding to individual community needs, rather than making overarching Western assumptions about what is needed?

 Line 120 – can the authors expand on why it is important for the geographical setting to be reported in the results as well as the methods section. This is not clear, particularly as we don’t know what exactly the CET includes or considers

P5 para 1 – is it that significant difference cannot be seen, or that the nuances and differences are not always reported. Difficulty in access may be significant, but look quite different, in urban, rural and remote areas.

Conclusion

Here the concept of Indigenous Governance is introduced. It needs referencing here and/or in the introduction. The CREATE tool also focuses on Indigenous leadership and governance, choice of methodology etc.

Line 135 – does it assure, or does it make it more likely? Without seeing/accessing the details of the CET tool it is difficult for the reader/reviewer to assess if this is a reasonable conclusion.

References

Also include references for CASP, EPHPP and CET tools

Ref 4 – author details are displaying incorrectly

Author Response

Section

Reviewer comment

Author response

General

The CET tool – exactly what is it? What does it contain? Did this team modify it or was it already modified?

Thank you for raising this, we have clarified this in lines:66-69

Suggested changes, addition of references and deeper discussion about terminology used – for example how can concepts of cultural safety, community settings, and reliability best be used together?

Thanks for your suggestions. In this paper, the CET is proposed as an addition to typical tools such as EPHPP and CASP. This approach sits alongside the more comprehensive CREATE tool, and complements efforts to decolonize research methods.

Underlying all of this research, the tools and appraisal, is a background of colonisation and decolonisation of Western health systems, health care and the way research is conducted and reported. Each country is in a different phase of addressing these aspects, and so identifying the country of origin of the papers and the tools is also important, particularly in an international context.

Thank you for your observations. The original systematic review only included literature from high income colonized countries. We agree about the importance of decolonizing research, which is why we have taken explicit steps in this research to decolonize the processes for selecting high value research to inform healthcare programs. We have added a sentence to address this.

Lines: 78-79

Title

This paper focuses more on the comparison of study quality, than the topic of Breast cancer – I suggest changing the order of the two parts of the title to reflect this

Comparison of study quality as determined by 3 standard research and community engagement metrics in breast cancer research in urban, rural and remote indigenous 2 communities.

Thank you for this suggestion, we have changed the order of the title to better reflect the focus of the study.

Abstract and main body

Use of word ‘typical’ to describe the tools. I suggest using another word such as ‘often’.  This enables a deeper consideration of whether the ‘typical’ tools used by Western research are appropriate and inclusive of Indigenous preferred approaches and priorities. ‘Typical’ seems exclusive of these considerations.

Thank you for this comment. We used the term ‘typical’ deliberately because the tools are exclusive of those considerations yet are presented as universal. Our paper is seeking to disrupt that idea.

The modified Indigenous community engagement tool – suggest put (CET) following in the abstract as this is used throughout the article.

Also did you modify this tool, or did you use an already modified version? This is unclear.

Added, thanks.

CET clarified, as above

Lines: 66-69

Reliability – this term is used throughout the paper. Can the authors explore/discuss more deeply the link and considerations between reliability as a quality measure, and recognising and responding to individual community needs and context as described in cultural safety. 

Also, are Western quant methods ‘reliable’ from an Indigenous community perspective?

To what extent have Aboriginal/Indigenous peoples been involved in critically reviewing the underlying assumptions and therefore the ‘reliability’ of these tools, from their perspective?

This is an emerging discussion amongst many Indigenous and non-Indigenous peoples internationally.

Thank you for the comment.

We are seeking to disrupt the notion of reliability as it applies to western methodologies and Indigenous research.

The authors use cultural safety as a concept throughout, but have not offered a reference to identify which definition of cultural safety they are referring to. Please include.

Thanks for pointing this out, we have added a sentence to clarify.

Lines: 56-59

There also seems to be a link being made between cultural safety and community settings and at times they are almost used interchangeably – Table 1 identifies setting, but nothing else about cultural safety - can this be explained more clearly

Thank you for this observation, we hope by clarifying cultural safety, we have separated out the ideas of cultural safety and community setting.

Tools

The CET tool – I am unclear exactly what the CET is. Having looked at reference 4 – NHMRC I am still confused as to exactly what the CET tool is, the best reference to find it, and whether this authorship team have modified it, or whether it has already been modified. This makes it difficult to assess and understand more deeply the connections being made between cultural safety, setting, and cultural engagement. Also whether Indigenous Governance is part of this assessment.

Thanks for the comment, we have endeavoured to address above.  

The importance of Indigenous Governance is introduced in the discussion section. I suggest also include in the Introduction (briefly).

Thanks for this suggestion, we have added to the introduction.

Lines 68-69

Suggest authors use a reference for each tool that enables the reader to go straight to the tool in question

Added references, with thanks

Also include the CET tool as a figure or table. Without this, it is hard to really review this paper and the discussion and conclusions effectively.

Please refer to our referenced Systematic Review, Table 2

These tools are also from different countries, please identify country of origin.  There may be contextual history and colonisation/decolonisation context behind each tool.

Thanks, we have added the country of origin.

Lines 74 and 76

The authors may also be interested in looking at the CREATE tool which is highly relevant. https://bmcmedresmethodol.biomedcentral.com/articles/10.1186/s12874-020-00959-3

I am not suggesting that you need to use this tool for this review and paper, (being that you have already completed the review) but you may find it useful in your next work. We are finding it increasingly useful, as it brings together many of the concepts you are discussing in this paper.

Objective: To develop an instrument that appraises the ethical and methodological quality of research conducted with Aboriginal and Torres Strait Islander peoples from an Aboriginal and Torres Strait Islander perspective.

The instrument is designed to be use used in conjunction with existing critical appraisal tools.

Noted with thanks. When we started our Systematic Review, the CREATE tool had not yet been published.

Introduction

P1 line 36 –38 “We assume that”…suggest rephrasing, it is unclear whether this is your assumption, or whether you are questioning this assumption (hopefully the latter). This assumption is pretty colonial if using Western tools.

Thanks for this, we have reworded

Lines: 46-47

P2 line 46 – this is where the reference clearly identifying the CET tool origin is needed.

The importance of Indigenous Governance is discussed for the first time in the discussion. It should be introduced here also.

Thanks, addressed above

P2 line 48 – ‘Typical’ suggest Western or similar term- as discussed in my terminology comments.

Thanks, addressed above.

P2 line 49- cultural safety – please provided some examples of what you mean by cultural safety and/or a reference

Thanks, addressed above

P2 –line 54 – a different approach is needed, rather than may be. Suggest you could use stronger language.

Thanks for the suggestions, we have changed the wording. Lines: 78-79

Method

Suggest rewording first sentence. We reassessed the studies selected…[or similar wording]

Thanks for the suggestion, we have changed the wording. Line: 89

It’s good that you have included that the authorship includes one Aboriginal author. This information is important.

Thanks, we agree

Results

Tables

The tables could benefit by some formatting/structural changes. It may also be that they were harder to read spanning across pages.

Thanks for the observation and noted

Headings- use same terminology for tools in all 3 tables to reduce confusion to readers new to the terms

Done, thanks

Table 1 – ‘Typical’ tools – do you mean CASP/EPHPP – if so, use those terms

Changed, thanks

Table 3- suggest used CET/ CADSP or EPIHPP

Changed, thanks.

Table 1 – suggest format each reference detail to left to make it easier to read. I note that these references are not included in the reference list, nor is the date of publication of these papers.   Perhaps include dates or include in reference lists, and/or refer to your original paper.  The dates are relevant as papers tend to improve quality over time. It may also be helpful to indicate which country each paper is from, as there may be trends as different countries respond to quality in writing Indigenous papers.

Add as references in table

Discussion

Line 99 – the value of research – to whom?

Value to Indigenous peoples. Lines:144-145

What does a cultural evaluation of reliability mean or look like? Is it that the study reliably works with community members? Or is reliability referring to quantitative studies?

Thank you for these questions. The cultural evaluation is intended to assess reliability through a cultural lens.

Line 103 Research method precision – what does this mean for the qualitative studies? Or is that sentence referring to quant and then qual studies?

Thanks for this, we have removed ‘precision’

Second paragraph – can the authors unpack a little more the generalizable findings, that are reliable through a cultural and community lens, and methodologies that are repeatable, when cultural safety is also about responding to individual community needs, rather than making overarching Western assumptions about what is needed?

Thank you for pointing this out. We agree it is confusing and have removed the sentence

Line 120 – can the authors expand on why it is important for the geographical setting to be reported in the results as well as the methods section. This is not clear, particularly as we don’t know what exactly the CET includes or considers

Thank you for the questions.

As stated in the discussion, whilst most papers collected information about geographical setting, none reported outcomes by setting. Geography is highly relevant to healthcare outcomes.

P5 para 1 – is it that significant difference cannot be seen, or that the nuances and differences are not always reported.

Difficulty in access may be significant, but look quite different, in urban, rural and remote areas.

Thank you for the question, we think it is both

We agree.  

Conclusion

Here the concept of Indigenous Governance is introduced. It needs referencing here and/or in the introduction. The CREATE tool also focuses on Indigenous leadership and governance, choice of methodology etc.

Thank you for pointing this out; we have endeavored to address above.

Line 135 – does it assure, or does it make it more likely? Without seeing/accessing the details of the CET tool it is difficult for the reader/reviewer to assess if this is a reasonable conclusion.

Thank you for this comment, we have changed the wording. Line:190

References

Also include references for CASP, EPHPP and CET tools

Done

Ref 4 – author details are displaying incorrectly

Done